# Synergistic Use of All-Acceptor Strategies for the Preparation of an Organic Semiconductor and the Realization of High Electron Transport Properties in Organic Field-Effect Transistors

**DOI:** 10.3390/polym15163392

**Published:** 2023-08-13

**Authors:** Shiwei Ren, Wenqing Zhang, Zhuoer Wang, Abderrahim Yassar, Zhiting Liao, Zhengran Yi

**Affiliations:** 1Zhuhai Fudan Innovation Institute, Guangdong-Macao Deep-Cooperation Zone of Hengqin, Zhuhai 519001, China; zhitingliao@126.com; 2Department of Materials Science, Fudan University, Shanghai 200433, China; 3Key Laboratory of Organic Solids, Institute of Chemistry, Chinese Academy of Sciences, Beijing 100190, China; zhangwq@iccas.ac.cn; 4Key Laboratory of Colloid and Interface Chemistry of Ministry of Education School of Chemistry and Chemical Engineering, Shandong University, Jinan 250100, China; wangzhuoer94@163.com; 5Laboratory of Physics of Interfaces and Thin Films-CNRS, Ecole Polytechnique, Institut Polytechnique de Paris, 91128 Palaiseau, France; abderrahim.yassar@polytechnique.edu

**Keywords:** n-type materials, organic semiconductor, conjugated copolymer, Stille coupling, OFET, electron mobility

## Abstract

The development of n-type organic semiconductor materials for transporting electrons as part of logic circuits is equally important to the development of p-type materials for transporting holes. Currently, progress in research on n-type materials is relatively backward, and the number of polymers with high electron mobility is limited. As the core component of the organic field-effect transistor (OFET), the rational design and judicious selection of the structure of organic semiconductor materials are crucial to enhance the performance of devices. A novel conjugated copolymer with an all-acceptor structure was synthesized based on an effective chemical structure modification and design strategy. PDPPTT-2Tz was obtained by the Stille coupling of the DPPTT monomer with 2Tz-SnMe_3_, which features high molecular weight and thermal stability. The low-lying lowest unoccupied molecular orbital (LUMO) energy level of the copolymer was attributed to the introduction of electron-deficient bithiazole. DFT calculations revealed that this material is highly planar. The effect of modulation from a donor–acceptor to acceptor–acceptor structure on the improvement of electron mobility was significant, which showed a maximum value of 1.29 cm^2^ V^−1^ s^−1^ and an average value of 0.81 cm^2^ V^−1^ s^−1^ for electron mobility in BGBC-based OFET devices. Our results demonstrate that DPP-based polymers can be used not only as excellent p-type materials but also as promising n-type materials.

## 1. Introduction

Since the traditional concept that organic compounds can only be used as insulating materials has been broken, the field of organic electronics has been extensively studied and rapidly developed over the past few decades [1]. The most fundamental research object in this field is organic semiconductor materials, which refer to organic small molecules and oligomer and polymer materials with π-conjugated architecture [2,3,4]. Organic materials are available from a wide variety of sources, with tailorable chemical structures, modifiable functional groups, and high flexibility. Through rational chemical design and molecular structure strategy modulation, the organic materials prepared have been applied in a variety of functional devices, such as the organic field-effect transistor (OFET), organic light-emitting diode (OLED), organic photovoltaic (OPV), organic light-emitting transistor (OLET), organic electrochemical transistor (OECT), and organic thermoelectric (TE) and so on [5,6,7,8,9,10]. Among them, OFETs are also organic thin-film transistors (OTFTs), which are regarded as the basic building blocks of organic electronic devices for carrying the function of transporting electrons or holes [11]. Electron-transporting materials are categorized as n-type materials, while p-type materials transport holes and materials that satisfy the need to transport both electrons and holes are referred to as ambipolar semiconductors. The current literature on organic semiconductors has focused on the high performance of p-type materials, while the development of electron transport materials has lagged behind. A variety of systems have been developed with hole mobilities above 10 cm^2^ V^−1^ s^−1^, while ambipolar or n-type polymer semiconductors with electron mobilities above 10 cm^2^ V^−1^ s^−1^ are quite rare. The primary reason for such a problem is the limited diversity of materials due to the low variety of acceptors. Common organic semiconductors with high LUMO energy levels have large electron–injection barriers. The diversity of available acceptor choices is limited, with most n-type materials being based only on systems consisting of isoindgio (IID), naphthalene diimide (NDI), perylene diimide (PDI), bithiophene imide (BTI), benzothiadiazo (BT) and diketopyrrolopyrrole (DPP) as acceptor units [12,13,14,15,16,17,18,19,20,21]. Further, air sensitivity is a particular issue in the development of n-type OSCs. The presence of electron traps significantly inhibits the performance of n-type devices. The fact is that the advancement of n-type organic semiconductors can contribute to a wide variety of smart applications. Notably, some unique areas, such as organic logic circuits, new integrated display technologies, and organic thermoelectronics, require high mobility and high-performance n-type organic semiconductors.

It is generally believed that a LUMO energy level near −4.0 eV favors the stability of the material in the air. It is relatively easy and efficient to achieve this energy level requirement using chemical strategies. It is beneficial to address the effect of molecular structure design strategies on material properties exemplified by DPP. As a weak acceptor structural unit, it can be polymerized with donor units such as thiophene, di-thiophene, or tri-thiophene to obtain polymeric materials with alternating donor–acceptor (D–A) arrangements, which correspond to hole mobilities in the range of 0.04–3.46 cm^2^ V^−1^ s^−1^ [13]. The chemical modification of the donor material to introduce strong electron-withdrawing groups such as –F atoms, –Cl atoms, or cyano contributes to lowering the LUMO energy level of the material, thereby increasing the electron mobility of the material. The copolymer of DPP combined with 3,4-difluorothiophe shows a 0.22 cm^2^ V^−1^ s^−1^ hole mobility and 0.19 cm^2^ V^−1^ s^−1^ electron mobility [16,22]. Lowering the proportion of donor motifs in the system, in other words increasing the proportion of acceptors in the polymer, means that the obtained P2DPP-based polymers exhibit ambipolar performance with maximum electron and hole mobilities of 3.01 cm^2^ V^−1^ s^−1^ and 4.16 cm^2^ V^−1^ s^−1^, respectively [23]. Further, the three-acceptor strategy was validated, and the P2DPP-BT-based polymers exhibited hole and electron mobilities of 3.52 cm^2^ V^−1^ s^−1^ and 2.83 cm^2^ V^−1^ s^−1^, respectively. Similarly, other acceptor-based building blocks are mostly used in a combined donor–acceptor mode, and the chemical structure of the donor unit is continuously modified to control LUMO energy levels and the performance of the material. Di–receptor and tri–acceptor strategies are currently effective strategies to enhance the device performance of n-type materials. Here, we propose a strategy for the structural design of all-acceptor (A–A) copolymers. We envision that the polymerization of electron-deficient bithiazole (2Tz) as the acceptor unit with the DPPTT acceptor block could significantly improve the electron transport properties of the material, prompting the conversion of the material from the p-type of PDPPTT-2T to the n-type of PDPPTT-2Tz (Figure 1). This design principle is in line with previous reports that bithiazoles lead to considerably higher ionization potentials and electron affinities compared to electron-rich bithiophene (2T) [24].

## 2. Materials and Methods

Materials and synthesis: Chemical reagents, organic solvents, and catalysts were purchased from Aldrich and used as received. Specific synthetic pathways for monomer-based materials **1**–**3** are described in the Appendix A. Compound **3** (200 mg, 0.18 mmol), 2Tz-SnMe_3_ (87.88 mg, 0.18 mmol 1.0 eq), tris(dibenzylideneacetone)dipalladium ([Pd_2_(dba)_3_], 3.24 mg, 3.54 μmol), tri(o-tolyl) phosphine (P(o-tol)_3_, 4.31 mg, 14.18 μmol), and anhydrous chlorobenzene (CB, 9 mL) was added to a Schlenk tube. This tube was charged with argon through a freeze-pump-thaw cycle three times. The reaction was polymerized at 130 °C for three days. After the polymerization was complete, sodium diethyldithiocarbamate trihydrate was added to remove [Pd_2_(dba)_3_], and the mixture was stirred for 30 min at 60 °C before being precipitated into methanol (350 mL). The precipitated product was filtered and purified via Soxhlet extraction with hexane (12 h), methanol (12 h), and acetone (12 h) to give fractions of blue, light red, and blue-violet colors. Finally, the target polymer was obtained using the chloroform phase, the fractions of which were dissolved in chloroform showing a dark green color. This fraction was concentrated by evaporation, precipitated into methanol (100 mL), and filtered to obtain the polymeric material PDPPTT-2Tz (184.4 mg, 91.7%) in the form of a dark powder.

Instrumentation: Nuclear magnetic resonance spectra recorded on a Bruker AVANCEIII (400 MHz, German) spectrometer with deuterated chloroform were used as the solvent. ^1^HNMR chemical shifts were referenced relative to internal tetramethylsilane. The splitting patterns were designated as s (singlet); d (doublet); t (triplet); and m (multiplet). Molecular weight was determined by gel permeation chromatography (Agilent PL-GPC 50, United States of America), utilizing 1,2,4-trichlorobenzene as an eluent. Mass spectrometry was performed on an Autoflex III (Bruker Daltonics Inc, German) MALDI-TOF spectrometer. The elemental analysis was measured by an organic elemental analyzer (Thermo Scientific Flash 2000, USA). UV-Vis spectra were recorded on a Varian Cary model 500 UV-Vis-NIR spectrophotometer (Agilent, USA) using standard quartz cells of 1 cm width and solvents of a spectroscopic grade. Electrochemical measurements were carried out in an acetonitrile solution with tetra-n-butylammonium hexafluorophosphate using a Metrohm Autolab PGSTAT12 Potentiostat. An Ag/AgCl electrode and platinum wire were used as the reference electrode and counter electrode, respectively. Density Functional Theory (DFT) calculations were performed using the B3LYP-D3/def2tzvp basis set of the Gaussian 16 program to elucidate the highest occupied molecular orbital (HOMO) and LUMO levels after optimizing low-energy conformation using the same method. Fourier transform infrared measurements were performed on a Thermo Scientific iN10 spectrometer. Thermogravimetric and Differential Scanning Calorimetry measurements were performed on a Mettler TGA/DSC thermal analysis system with a heating rate of 10 °C min^−1^ in a nitrogen atmosphere.

Device: The substrates for polymer-based OFETs were subjected to cleaning using ultrasonication in deionized water, acetone, and isopropanol. The cleaned substrates were dried under nitrogen gas and then treated with plasma for 10 min. Before the deposition of polymer semiconductors, octadecyltrichlorosilane (OTS) treatment was performed on the SiO_2_ gate dielectrics in a vacuum to form an OTS self-assembled monolayer. The field-effect characteristics of the devices were determined in the glove box by using a 4200 SCS semiconductor parameter analyzer (Keithley, China). Different channel lengths (L) of the FET devices (L = 5, 10, 20, 30, 40, and 50 µm) and the same channel widths (W) of 1400 µm were used to optimize device performance.

## 3. Results

### 3.1. Synthesis

**PDPPTT-2Tz** was synthesized following the procedure summarized in Figure 2. The long side chain was introduced at the N-position of **1** through an alkylation reaction to generate intermediate **2** and enhance its solubility in organic solvents. Bromination under heating conditions was utilized to minimize the reaction time, whereby N-bromosuccinimide was employed to afford **3**, which could serve as a precursor for the Stille coupling reaction. Polymerization occurred via the dibromo-DPPTT monomer (**3**) and 5,5′-bis(trimethylstannyl)-2,2′-bithiazole (2Tz-SnMe_3_**)** with palladium (0) as the catalyst. It is worth noting that polymerization can often be switched on within ten minutes using stoichiometric equivalents of catalysts and ligands. An obvious experimental phenomenon is that within ten minutes from room temperature to 130 °C, the color of the solution can change from red to purple and blue to dark green. The dark green color is a sign that the polymerization of DPP-type polymers has started. The purification of target polymers relies mainly on the Soxhlet extraction technique. Hexane, methanol, and acetone were sequentially used to remove oligomers and low molecular weight fractions from the polymer system, and high molecular weight macromolecular fractions were collected by the chloroform solvent, resulting in a 91.7% yield. As mentioned earlier, the choice of these two monomers is favorable for device performance. Firstly, Br-containing and Sn-containing monomers can easily undergo the Stille coupling polymerization reaction, which reduces the difficulty of material generation. At the same time, its ability to draw electrons as an electron acceptor can significantly reduce the LUMO energy level of the material. The material energy level matches more closely with the Au electrode energy level, enabling the easy injection of electrons. Finally, both acceptor units have a planar structure, which is favorable for electron mobility.

High-temperature GPC tests (150 °C) showed that the polymer has an extremely high molecular weight with a narrow polymer dispersibility index (PDI). The Mn, Mw, and PDI of **PDPPTT-2Tz** exhibited 171.2 kDa, 257.8 kDa, and 1.50, respectively (Appendix A). For the mostly DPP-based polymers whose molecular weights (Mn) were generally less than 100 kg mol^−1^, we attributed this to the minimal use of catalysts with reasonable ligand ratios combined with long polymerization times [25,26,27]. In general, the molecular weight of the polymer is related to device performance [28,29]. High molecular weight components predominantly adopt a planar π-stacking conformation, which allows for efficient interchain charge carrier transport [30]. Along with this, the narrow distribution favors carrier transport compared to a high PDI [31]. On the other hand, the higher molecular weight decreased the solubility of the polymer, which was almost insoluble in non-chlorinated solvents such as tetrahydrofuran, ethyl acetate, ethanol, etc [32]. The solubility of the polymer in chloroform and chlorobenzene was 3.5 mg/mL and 7.5 mg/mL, respectively, when heated to 50 °C. The reason for using chlorobenzene as a solvent at the beginning of the reaction was designed to avoid the precipitation of high molecular weight polymers with reduced solubility during the reaction. Thermogravimetric analysis (TGA) indicated that the polymer had excellent thermal stability, with a thermal decomposition temperature (5% weight loss) of 416.5 °C in the nitrogen atmosphere (Figure 1). Differential scanning calorimetry analysis indicated that there was no significant phase transition in the temperature range of 300–350 °C (Appendix A), which could be attributed to the rigid skeleton and high molecular weight of **PDPPTT-2Tz**. Due to the poor solubility of polymers, the NMR test could not be used to characterize the copolymer structure. We, therefore, used infrared spectroscopy to compare the difference in absorption between the monomers and polymers (Appendix A). The two materials presented completely different peaks in the wavenumber range of 600–4000 cm^−1^, with the typical carbonyl characteristic peak migrating finely from symmetry at 1672 cm^−1^ to 1665 cm^−1^ and showing asymmetry. The new peaks at around 850 cm^−1^ as well as at around 3500 cm^−1^ were attributed to the insertion of the thiazole unit. The elemental analysis of polymers is another means of testing their composition and purity. The data and average values of the two measurements for the C, H, N, and S content of these molecules are in good agreement with the theoretical values calculated from the composition of repeating units (Appendix A). These indicate that the material is of high purity and that impurities, including catalysts and ligands, have been purified out.

### 3.2. Density Functional Theory Calculations

The optimized molecular geometries, highest occupied molecular orbital (HOMO) distribution, lowest occupied molecular orbital (LUMO), and the energy gap (ΔEc) were calculated using DFT at the B3LYP-D3/def2tzvp level [33,34] on the basis of the Gaussian 16 program [35,36]. The low-energy conformation of the methyl-substituted trimers was applied instead of the long-conjugated system of the polymer [37]. Notably, the effect of adding dispersion correction to the theoretical calculations was intended to further precise the intramolecular forces and energy levels of the material with only a minor additional computational overhead [38,39]. The smaller thiophene unit only had one α-hydrogen atom, which allowed for favorable intramolecular sulfur–oxygen interactions with a calculated S…O distance of 2.98 Å, which was less than the sum of the van der Waals radii of 3.32 Å [40]. The central backbone is oriented in a nearly coplanar manner with its neighboring thiophene units at a dihedral angle of 10.17° to each other (Figure 2a). The introduction of the 2Tz group did not significantly affect the planarity of the main chain structure. This is due, on the one hand, to the fact that 2Tz itself consists of two thiazole units linked in a trans conformation with a dihedral angle between the thiazole rings close to 180°. On the other hand, the dihedral angle between the thiophene unit and the neighboring thiazole in DPP is only 4.95° (Figure 2b). The calculated LUMO energy level was −3.79 eV, and the lower energy level coincided with the all-acceptor molecular structure design strategy (Table 1). Thereby, it allowed for the easier electron injection and operational stability of the resulting polymer. The energy gap (ΔE_g_) obtained from theoretical calculations was 1.81 eV. Non-covalent interaction scattering and reduced density gradient analyses were performed to observe and differentiate non-covalent interactions within the molecule, as shown in Figure 2c [41]. Theoretical simulations of Electrostatic potential surfaces and the molecular orbital maps of the trimers are shown in Appendix A, respectively.

### 3.3. Photophysical Properties and Electrochemical Properties

The UV-visible absorption spectra of **PDPPTT-2Tz** in the solution and in thin films are illustrated in Figure 3 below, and the spectral characteristics are summarized in Table 2. Two absorption peaks could be seen in the solution, including a high-energy absorption peak in the range of 350–500 nm and a low-energy absorption peak in the range of 600–900 nm, which were attributed to the π–π* transition and the intramolecular charge transfer transition, respectively (Figure 3a). The maximum absorption peak (λ_max_) of copolymer **PDPPTT-2Tz** in the chloroform solution was 765 nm. Compared to the solution absorption, the λ_max_ of the film exhibited a pronounced redshift of 10 nm. The changes observed in the absorption spectra from the solution to the solid state were the result of enhanced intermolecular interactions and increased molecular ordering levels. J aggregation could be determined by observing the difference in the intensity ratios of the first two vibrational peaks (often referred to as the 0–0 and 0–1 transitions) in the absorption spectrum. The films show an optical band gap (E_g_^opt^) of approximately 1.31 eV, as determined from the onset of UV-visible absorption at approximately 946 nm.

The redox potential of the **PDPPTT-2Tz** film was investigated using cyclic voltammetry (CV) in a 0.1 M TBAPF_6_ solution in anhydrous acetonitrile. The polymer film was prepared as follows. Firstly, 1 mg of the sample was dissolved in 1 mL of CB, and 0.5 mL of the above solution was aspirated with a needle. In total, 1–2 drops were placed on the electrode while waiting for the reagent to evaporate. The test conditions were measured at room temperature in a nitrogen atmosphere. The reduction peaks were shown to be quasi-reversible, which was attributed to the carbonyl group within the skeleton and was progressively reduced. On the other hand, the oxidation process showed a significantly lower and less reversible current, suggesting that **PDPPTT-2Tz** could favor electron transport rather than hole conduction. The HOMO level was calculated using the onset of the oxidation potential measured by CV, and the LUMO of the polymers was calculated using the onset reduction potential with an offset of −4.80 eV for saturated Ag/AgCl (Figure 4). The ferrocene reference was obtained under the same test conditions as 0.39 V. We observed an onset oxidation potential of 0.67 V and an onset reduction potential of −0.69 V, corresponding to −5.08 eV for HOMO and −3.72 eV for LUMO (Table 2). The energy gap (E_g_^cv^) calculated from electrochemical measurements was 1.36 eV, which was close to the optical bandgap (E_g_^opt^). These low-lying LUMO energy levels are consistent with the DFT calculations, indicating that the all-acceptor strategy effectively decreased the energy levels of the material.

### 3.4. OFET Performance

In order to investigate the charge transfer characteristics of **PDPPTT-2Tz**, we fabricated OFET devices with a bottom–gate bottom–contact (BGBC) structure, where the corresponding BGBC device configuration is shown in Figure 5a. Polymer-based OFETs were fabricated on a highly doped silicon wafer with a 300 nm SiO_2_ insulator, which was used as the gate electrode. The source-drain gold electrodes were formed by photolithography. Then, a layer of the polymer semiconductor film was deposited on the OTS-treated substrates by spin-coating from the polymer solution in hot o-dichlorobenzene (5 mg/mL) at a speed of 2000 rpm for 60 s. For annealing the semiconductor film, the samples were further placed on a hotplate in a glove box at 220 °C for 20 min before cooling down to room temperature. The carrier mobility, threshold voltage, and on/off current ratio extracted from the transfer characteristics are summarized in Table 3. The field-effect mobility in saturation (µ) was calculated from the following equation:I_DS_ = (W/2L) C_i_ µ (V_GS_–V_th_)^2^(1)
where W/L is the channel width/length, C_i_ is the gate dielectric layer capacitance per unit area, and V_GS_ and V_th_ are the gate voltage and threshold voltage, respectively.

As shown in Figure 5b, it is clear to see that the **PDPPTT-2Tz**-based OFET device showed N-type dominant carrier transfer characteristics with a maximum μ_e_ value of 1.29 cm^2^ V^−1^ s^−1^ with a channel length of 40 µm. The maximum possible difference between the I_on_ and I_off_ states was > 10^6^, which was calculated between the maximum gate bias (+80 V) and the minimum gate bias just below the switching voltage, showing that the material had an excellent switching property [42,43]. The average μ_e_ value of 0.81 cm^2^ V^−1^ s^−1^ relatively outperformed N-type devices based on DPP-based polymers [44,45]. On the other hand, the hole mobility was extremely low in the range of 0.003–0.007 cm^2^ V^−1^ s^−1^, which could be related to the nature of the system as electron deficient. The output characteristics of the **PDPPTT-2Tz**-based device are shown in Figure 5c,d.

## 4. Discussion

Here, we believe that the significant improvement in electron mobility was due to multiple contributing factors. Firstly, the reduction in the LUMO energy level due to the all-acceptor strategy is crucial, as this was favorable for tuning the energy bands of the semiconductor as well as for matching with the working electrode (Au). Secondly, the effect of molecular weight on mobility cannot be ignored, especially since the materials polymerized in this paper had extremely high molecular weights and narrow dispersions. The increase in the effective conjugation length was the key to improving carrier mobility. At the same time, a reasonable choice of acceptors resulted in a more planar structure of the overall molecule. It favored intermolecular stacking and π–π interactions, thus facilitating electron transport. Although carriers in the range of 1.0 cm^2^ V^−1^ s^−1^ are already sufficient for the functioning of most basic devices, such as e-paper, sensors, electronic labels, etc., there is still a gap with materials with electron mobility of more than 10 cm^2^ V^−1^ s^−1^, as reported in the literature. This is related to the structure and molecular design strategy of the material itself on the one hand and the preparation method of polymer semiconductor films on the other. The enhancement of carrier transport in polymer semiconductors by external forces conferred by rod coating, film scraping, spraying, and solution shearing was decisive for the performance of the devices. In addition, doping and blending might be other options for enhancing the device’s performance.

Research in this field for n-type organic semiconductor materials and their applications in functional devices still has some limitations and is a major direction for future research. Firstly, most polymers are based on Stille coupling polymerization, which requires the use of toxic Sn-containing reagents. On the one hand, this increases the cost of synthesis, and on the other hand, it can easily contaminate human beings and the environment. The current alternative is to use C–H activation for polymerization, which reduces or even eliminates the use of Sn reagents. However, this method also has the disadvantages of low yield, low molecular weight polymerization, and poor generalizability. Second, the reproducibility of the polymers is inferior, as the molecular weight or dispersion of the polymers cannot be exactly the same from each batch, which is insufficient compared to the purity of the small molecules. The optimization of coupling polymerization reaction conditions, the development of efficient green polymerization methods, and the use of mild and environmentally friendly solvents and catalysts are essential points to explore. Finally, n-type semiconductor transistor materials are currently tested in glove boxes, and it is difficult to prepare materials that are stable in the air for long periods of time with long lifetimes.

Realizing the stretchable properties of the material while satisfying the mobility could be the next major extension of this project. The intrinsic stretchability of materials is important because many electronic devices need to be stretchable and bendable in order to achieve a wide range of functionalities. Compared to small molecule semiconductors, polymer semiconductors have an advantage in the field of stretchability due to their longer and softer chain segments. In general, it is more difficult to balance the mechanical and electrical properties of materials. Most of the materials reported exhibit good electron mobility under unstressed conditions, but their electrical properties tended to degrade rapidly under 50% or greater stress. The PDPPTT-2Tz-based material possesses good electrical properties and can withstand the loss of some of its electron mobility while accommodating stretchable performance.

## 5. Conclusions

A novel organic semiconductor polymer, **PDPPTT-2Tz**, was designed based on an all-acceptor (A–A) structural strategy and was prepared by the palladium-catalyzed Stille polycondensation of electron-deficient 2Tz and DPPTT monomers. A polymer with high molecular weight and narrow dispersion can be prepared by the design of suitable experimental conditions. The enhancement of electron mobility by the A–A structure is significant compared to D–A structure polymers. DFT calculated results show that PDPPTT-2Tz has a lower LUMO energy level, which contributes to an improvement in its electron attraction. The two selected monomers with planar surfaces can still maintain good planarity when forming the polymer, given that the main chain has a tendency to have a large planar. The high molecular weight and planarity further favor its electron transport properties, and the polymer has been used in high-performance OFET devices showing good electron mobility and on-off ratios. Subsequent work on the stability of the device and further enhancement of the carrier mobility and its stretchable properties is still in progress.

## Data Availability

Not applicable.

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
