# Peer review of "Synergistic Use of All-Acceptor Strategies for the Preparation of an Organic Semiconductor and the Realization of High Electron Transport Properties in Organic Field-Effect Transistors"

_polymers, 2023, doi:10.3390/polym15163392_

Round 1
Reviewer 1 Report
The development of n-type organic semiconductors for electron transport in logic circuits is pivotal, yet lagging compared to p-type counterparts. A novel conjugated copolymer, PDPPTT-2Tz, with an all-acceptor structure was synthesized to enhance organic field-effect transistor (OFET) performance. Achieved through Stille coupling and enriched with electron-deficient bithiazole, the copolymer boasts high molecular weight, thermal stability, and a low-lying LUMO energy level. Its shift from donor-acceptor to acceptor-acceptor structure notably improved electron mobility, reaching up to 1.29 cm2 V−1 s−1. This highlights DPP-based polymers' potential as both p-type and promising n-type materials.
-
All-acceptor Strategies Emphasis: The uniqueness and importance of the "All-acceptor Strategies" is evident as the main selling point of this research. However, its coverage in the introduction appears inadequate, potentially leaving readers unfamiliar with this topic a bit in the dark. I recommend that the authors expand on this strategy in the introduction, providing a clearer background and justifying its significance in the field.
-
Correction in Section 3.1: It has been noted that in the 3.1 paragraph, the yield should be "91.7%". Please ensure that the numbers are accurate and consistent throughout the manuscript.
-
Figure S2 Annotation: In Figure S2, it would be beneficial for readers if the decomposition temperature could be explicitly marked. This would improve clarity and ease of interpretation.
-
Arrow Annotation in Figure 1a: I suggest adding arrows in Figure 1a to precisely indicate where specific values or features are being referenced. This would greatly enhance the figure's accessibility to the readers.
-
Clarification in Section 3.3: The statement in paragraph 3.3 implies that the red-shift from solution state to film state is solely due to π-conjugation lengthening.("further demonstrating the material's effec- tive π-conjugation lengthening and HOMO/LUMO coupling enhancement") However, as rightly pointed out, the aggregation-induced red-shift is a common phenomenon, irrespective of the π-conjugation lengthening, especially since the material remains the same. The authors should consider refining or elaborating on this point to avoid potential misunderstandings.
-
CV Measurements: For better accuracy and reproducibility of the cyclic voltammetry (CV) measurements, it would be beneficial to incorporate an internal standard such as ferrocene. This would provide a clearer point of reference.
-
Recent Literature Citations: The authors should consider updating the manuscript by citing some recent works in the field of OPV, DPP, and A-A DPP, namely:
- Adv. Funct. Mater.2023, 33, 2215095
- Polym J 55, 507–515 (2023). https://doi.org/10.1038/s41428-022-00717-w
Incorporating these studies will not only enhance the depth of the literature review but also provide readers with a more comprehensive understanding of the recent advancements in the field.
Reviewer 2 Report
Overall, the manuscript provides valuable insights into the development of n-type organic semiconductor materials and their potential use in logic circuits. However, there are some areas that could be improved to strengthen the manuscript:
-
Introduction and Context: Expand on the significance of n-type organic semiconductors in electronic devices and their challenges compared to p-type materials. Additionally, cite some recent relevant literature to support the claim that progress in n-type materials research is relatively backward.
-
Methodology: Provide a detailed description of the synthesis process for PDPPTT-2Tz, including reaction conditions, purification methods, and characterization techniques used to confirm the chemical structure. This will help readers to replicate the experiment and validate the results.
-
Discussion on Molecular Structure: Explain the rationale behind choosing the specific monomers and the role of 2Tz-SnMe3 in the synthesis. Elaborate on how the introduction of the electron-deficient bithiazole affects the LUMO energy level and subsequently improves electron mobility.
-
Experimental Data: Include more experimental data in the form of graphs, figures, or tables to support the claims made in the text. For example, show the DFT-calculated planar structure of the material and provide relevant characterization data like X-ray diffraction, SEM images, or thermal analysis results.
-
Comparative Analysis: Compare the performance of PDPPTT-2Tz with other existing n-type materials to highlight its unique features and advantages. Discuss the potential applications of the material in practical electronic devices and how it compares to state-of-the-art n-type organic semiconductors.
-
Discussion on Mobility Improvement: Provide a deeper discussion on the factors that contribute to the significant improvement in electron mobility due to the modulation from donor-acceptor to acceptor-acceptor structure. Is it solely attributed to the electron-deficient bithiazole, or are there other factors involved?
-
Limitations and Future Directions: Acknowledge any limitations in the study and propose potential future research directions to address these limitations. This will add credibility to the work and stimulate further investigations in the field.
By addressing these points, the manuscript will be more comprehensive, informative, and suitable for publication.
minor
